# The Protective Effects of Sesamin against Cyclophosphamide-Induced Nephrotoxicity through Modulation of Oxidative Stress, Inflammatory-Cytokines and Apoptosis in Rats

**DOI:** 10.3390/ijms231911615

**Published:** 2022-10-01

**Authors:** Saeed Alshahrani, Hani M. Ali Thubab, Abdulrahman M. Ali Zaeri, Tarique Anwer, Rayan A. Ahmed, Abdulmajeed M. Jali, Marwa Qadri, Yousra Nomier, Sivakumar S. Moni, Mohammad F. Alam

**Affiliations:** 1Department of Pharmacology and Toxicology, College of Pharmacy, Jazan University, Jazan 45142, Saudi Arabia; 2Department of Pharmaceutics, College of Pharmacy, Jazan University, Jazan 45142, Saudi Arabia

**Keywords:** cyclophosphamide, sesamin, nephroprotective, oxidative stress, inflammatory cytokines, apoptosis

## Abstract

Cyclophosphamide is an anticancer drug with a wide spectrum of clinical uses, but its typical side effects are multiple complications, including nephron toxicity. The possible molecular mechanism of the nephroprotective action of sesamin (SM) against cyclophosphamide (CP) induced renal toxicity was investigated in rats by understanding oxidative stress and inflammatory cytokines. In this study, rats were arbitrarily grouped into the following four groups: a normal control group (CNT); a CP-induced toxicity group; a treatment group with two doses of sesamin SM10 and SM20; a group with sesamin (SM20) alone. A single dose of CP (150 mg/kg body, i.p.) was administered on day 4 of the experiments, while treatment with SM was given orally for seven days from day 1. The group treated with SM showed a significant protective effect against CP-induced renal damage in rats. Treatment with SM significantly increased the antioxidant enzymes (GSH, CAT, and SOD) and reduced malondialdehyde (MDA) levels. Thus, SM significantly overcame the elevated kidney function markers (creatinine, blood urea nitrogen, and uric acid) by attenuating oxidative stress. The SM also significantly reduced the elevated cytokines (IL-1β and TNFα) and caspase-3 in the treated group. Histopathological studies confirmed the protective effect of sesamin (SM) on CP-induced nephrotoxicity. In conclusion, the current findings support the nephroprotective effect of sesamin against CP-induced renal injury.

## 1. Introduction

Chemotherapy is an effective treatment for various kinds of cancer diseases, but it causes multiple side effects. The side effects of chemotherapy are different for each patient depending on the type of cancer, location, drug, dose, and the patient’s immune system. Most of the drugs including anticancer and their metabolites cleared from the renal pathway and causes major damage to the kidney [1]. Cyclophosphamide is one of the anticancer drugs with a wide spectrum of clinical uses, such as sarcoma, lymphoma, neuroblastoma, multiple myeloma, leukemia, breast cancer, ovarian and lung cancer, etc. Its typical side effects are multiple complications, including nephrotoxicity [2].

The metabolic activation of CP by the hepatic enzyme is 4-hydroxy cyclophosphamide, which converts into two toxic metabolites such as scarlein and phosphoramide [3]. This metabolite makes a covalent bond with DNA and proteins, resulting in cell death. A small fraction of CP is removed by the kidney, while the tubular reabsorption of this drug is very high [4]. Various researchers have reported that oxidative stress plays an important role in CP-induced renal injury [5,6,7]. Researchers have reported that CP-induced reactive oxygen species (ROS) play a key role in the pathogenesis of CP-independent renal damage. In this way, ROS-induced oxidative stress in the proximal tubules of the kidney mainly affects the cells, leading to nephrotoxicity [8]. Thus, natural scavengers of free radicals and antioxidants can be used to treat kidney damage caused by CP. It is now possible to reduce or eliminate the toxic effects of cyclophosphamide by combining it with various detoxifying and protective agents [9].

Therefore, palliative care as free radical scavengers and antioxidants is always necessary for cancer treatment, including CP. Natural resources have been used as a therapeutic medicine for thousands of years. Nevertheless, the majority of the population of a developing country relies on herbal medicine for primary health care [10]. The seeds of sesame plants and their oil have been used in human diets for thousands of years as a healthy food source [11]. Sesamin is an active ingredient of sesame seeds (*Sesamum indicum*) and sesame oil known for various physiological actions. Sesamin is a major lignin that is absorbed proficiently and circulated throughout the body. It is highly distributed in the liver and kidney as sesamin metabolites and plays various physiological effects such as antioxidant [12], anti-inflammatory [13], a protective role against mitochondrial toxicity [14], a lowering effect on serum cholesterol, lipid level, and blood pressure [15,16], a protective effect against alcohol-induced liver injury [17] and anti-allergic effect [18], etc. A recent study published by B H Ali (2020) [19] indicated sesamin ameliorated cisplatin-induced nephrotoxicity, but cyclophosphamide-induced nephrotoxicity has not been explored yet. Therefore, this idea encourages us to explore the defending action of Sesamin against cyclophosphamide-induced nephrotoxicity.

## 2. Results

### 2.1. Kidney Function Test

Table 1 showed blood urea nitrogen (BUN), uric acid, and creatinine content that increased significantly (*p* < 0.001) in the CP administered group as compared to the normal control group. However, SM10 and SM20 treatment showed remarkable improvement and restoration in these active markers of the kidney. Significant improvement of these markers was seen in pretreatment with SM20 than SM10 as compared to CP administration, but no significant changes were seen in alone as positive control SM20 treatment vs. normal control.

### 2.2. Oxidative Stress Markers

Table 2 represents the MDA level that was remarkably amplified in the CP administration group than in the normal control group. The CP caused a significant decrease (*p* < 0.05) in reduced glutathione (GSH), catalase (CAT), and superoxide dismutase (SOD) content compared to the normal control group (CNT). Pre-treatment with SM10 and SM20 significantly improved these levels (GSH, CAT, and SOD) and reduced the level of MDA. It was also noticed that SM20 was more effective and highly significant than SM10. The administration of SM10 failed to increase the level of SOD as compared to CP administration. Pre-treatment with SM20 alone as a positive control did not find any significant changes in these markers.

### 2.3. Inflammation and Apoptosis

Figure 1 and Figure 2 showed a significant (*p* < 0.0001) increment in pro-inflammatory cytokines (IL-1β, and TNFα) after CP administration in comparison to CNT. In contrast, seven days of continuous pre-treatment with SM10 and SM20 drastically lowered these inflammatory cytokines in comparison to the CP alone administered group. However, these cytokines were not significant between SM20 alone and normal control groups. Figure 3 represents the effects of Sesamin on apoptosis markers. Interestingly, caspase 3 was increased significantly (*p* < 0.0001) in the CP administered group, while pre-treatment of SM10 and SM20 remarkably decreased the level of caspase 3 in the SM10 + CP and SM20 + CP groups. It was noticed that no significant difference was found between SM20 alone and CNT.

### 2.4. Histopathology

Figure 4B shows the harmful effects of cyclophosphamide (CP) on kidney histology in the CP treatment group in comparison to the normal control group (CNT). The impact of CP toxicities was very prompt with glomerulus nephritis, missing capillary loops, open Bowman’s space, edema, hemorrhage, and vacuolization, in the CP administered group than in CNT. However, pre-treatment with SM10 showed minor improvement in glomerulus morphology (Figure 4C), but remarkable regaining was observed in the SM20 group (Figure 4D) compared to CP alone. In the sesamin higher dose (SM20) treated group (Figure 4E), no lesions were observed. The normal control group (CNT) showed no sign of inflammation or necrosis (Figure 4A).

## 3. Discussion

Chemotherapeutic drugs improve cancer patient survival, but nephrotoxicity remains a critical complication and challenge. Cyclophosphamide is a chemotherapeutic drug that causes renal injuries at a different level that reflects in serum markers. In a previous report, cyclophosphamide therapy was reported to develop renal toxicity in cancer patients [20,21]. Current cancer therapeutic strategies are needed to avoid direct or indirect renal toxicity by using natural or complementary antioxidants. Sesamin was used in this study to explore the possible mechanism linked to antioxidant markers, inflammation markers, and apoptosis markers behind renal protection. Our findings showed that sesamin attenuates the CP-based nephrotoxicity in rats.

The progression of renal toxicity after CP administration is characterized by increases in serum BUN, uric acid, and creatinine [22,23,24]. The current outcomes showed a significant elevation in the BUN, uric acid, and creatinine in the CP administration group. The elevated serum markers may be due to renal damage, as evidenced by decreased creatinine clearance and low glomerular filtration rate. The measurement of these markers proves helpful in diagnosing renal toxicities [25].

In the past few decades, experimental animal research has shown that CP nephrotoxicity is linked with the free radical induced oxidative stress that mediates extensive renal impairments [26]. Cyclophosphamide causes renal toxicity by alkylation of the renal cell by the Cys sulfhydryl group of acrolein, one of CPs active toxic metabolites. As per scientific evidence, tubular reabsorption of CP is very high; therefore, a minimal amount is excreted from the kidneys [27]. CP experiences metabolic activation through liver enzymes and forms 4-hydroxycyclophosphamide that further converts into two metabolites, acrolein and phosphoramide mustard, which is cytotoxic [28]. Acrolein is an electrophilic and highly reactive aldehyde that interacts with the antioxidant defense system [29]. The acrolein can bind with reduced glutathione (GSH) and reduce its renal cell level. Thus, acrolein hampered the antioxidant system by producing more free radical generation, which is responsible for oxidative stress and damaging the cells of the kidney. The renal alkylation process reduces the glomerular filtration rate and causes tubular dysfunction [30].

The current observation indicates that a single dose of CP caused lipid peroxidation and MDA content were increased compared to CNT, while antioxidant enzymes were significantly reduced. These observations are supported by the previous investigator where CP produced lipid peroxidation and caused renal damage in rats [31,32]. Sesamin treatment remarkably restores these antioxidant defense systems in the present study. Sesamin has antioxidant [33], anti-inflammatory [34], and anti-apoptotic [35] properties, which are already proven. It has been suggested that sesamin’s metabolites (monocatechol and dicatechol) are responsible for various physiological effects. These metabolites exhibited antioxidant activity in the liver cells [36]. Sesamin was sequentially metabolized by cytochrome p450 and UDP glucuronosyltransferase or sulfotransferase. Sesamin metabolism was mainly mediated by CYP2C9 in the human liver [37]. A previous study indicates that sesamin suppresses increased NADPH oxidase-dependent oxidative stress in diabetic mice. NADPH inhibition may therefore result in various anti-inflammatory effects. As a result of oxidative stress, pro-inflammatory cytokines are produced as well as different signaling pathways are stimulated, resulting in inflammation, fibrosis, and apoptosis [38,39].

Inflammation shows a vital role in CP-involved renal pathogenesis by involving several molecular and cellular factors. As a result of CP-induced oxidative stress, the transcription factor NF-κB is activated in the nucleus, promoting a series of inflammatory changes and IL-1 and TNF-expression [40]. Thus, increased concentration of pro-inflammatory cytokines leads to activation of NF-κB, which results in increased transcription of the pro-apoptotic genes and eventually causes apoptotic cell death [41].

In the present study, these cytokines (IL-1β and TNFα) levels were increased in the kidneys of rats after exposure to CP, and their effects were significantly inhibited by the treatment with Sesamin. In addition, pro-inflammatory cytokines are also responsible for activating the p38-MAPK pathway that acts as a serious role in cellular apoptosis [42,43]. Apoptosis is programmed cell death associated with the activation of caspases, which is called the initiator or executioner of apoptosis [44].

Caspase 3 plays an important role in apoptosis in renal tubules [45,46]. This study found that caspase 3 levels increased after CP exposure and decreased after sesamin treatment.

In addition to the biochemical findings discussed previously, histopathological findings confirm that sesamin can attenuate renal injury induced by CP. In the presence of CP, infiltration of inflammatory cells and significant changes in renal architecture were noted, but sesamin ameliorated the effect and improved renal structure and function. Overall, the protective and antioxidant properties of sesamin contribute to the normal histological architecture of renal tissue.

## 4. Materials and Methods

### 4.1. Chemicals

Sesamin, cyclophosphamide, 5,5dithiobis-(2-nitrobenzoic acid), reduced GSH, TBA, were obtained from Sigma chemicals (3300 S 2nd St #3306 St. Louis, MO 63118, USA). Biochemical assay kits for BUN, Uric Acid, and Creatinine were obtained from Randox, Crumlin, UK. Elisa estimation kits for IL-1β, TNFα, and Caspase-3 were purchased from Abcam, Cambridge, UK.

### 4.2. Animals

Male Wistar rats weighing 150–160 g purchased from the MRC, Jazan University, Saudi Arabia. The rats were adapted under normal laboratory conditions of 12 h dark and light cycle, temperature (20 ± 2 °C), and humidity (50 ± 15%). The animal experiments were carried out after ethical approval no REC41/1-032 from the Research Ethical Committee, Jazan University.

### 4.3. Experimental Design

The rats were grouped into five test groups having six rats in each. Normal control (CNT) received the same number of vehicles as saline only. Test groups were further classified into four groups as rats received a single dose of Cyclophosphamide with saline i.p. on day four, represented as CP alone. The third and fourth groups of rats received two oral doses of sesamin (SM10 and 20 mg/kg) [47] from day one to day seven along with a single dose of CP (150 mg/kg) on day four, represented as SM10 + CP and SM20 + CP, for seven days respectively. Another group of rats received only sesamin (SM20 mg/kg) p.o. daily for seven days. Rats of all groups except normal and sesamin control were used as single dose administration intraperitoneally of CP (150 mg/kg body weight) [48] on day four.

After seven days of treatment (i.e., on day 8), rats fasted for a minimum of 12 h. All rats of each group were anesthetized with chloral hydrate, and blood was collected immediately from them through the retro-orbital plexus. After that, the animal was sacrificed, and kidneys were removed from each rat and divided into two parts. The first parts were kept at −80 °C for tissue preparation and biochemical analysis. In contrast, another part of the kidney was immersed in a 10% formalin solution for histological investigation by a light microscope. Samples of blood were centrifuged at 300 rpm for 15 min; serum was extracted and stored at −20 °C for kidney function tests. The tissue was homogenized in 10% child PB solution (0.1 M and pH 7.4). The homogenized tissue was centrifuged at 3000 rpm for 15 min at 4 °C, and the upper layer was used to estimate oxidative stress and cytokine assay.

### 4.4. Assessment of Renal Function Test in Serum

Determination of renal function tests (blood urea nitrogen, uric acid, and creatinine) was carried out as per the standard procedure of Randox UK using a photometer.

### 4.5. Assessment of Oxidative Stress Markers

Assessment of malondialdehyde (MDA) in renal tissue was carried out by Ohkawa et al. (1979). In a nutshell, the mixture contained 1.5 mL of 20% glacial acetic acid, 1.5 mL of aqueous solution of TBA, 0.2 mL of 8% SDS, and 0.8 mL of sample. This mixture was further heated at 95 °C for 60 min. After being cooled with tap water, the solution was agitated vigorously before 5 mL of a 15:1 butanol and pyridine combination and 1 mL of distilled water were added. The mixture was then centrifuged at 2000× *g* for 10 min, and the supernatant’s absorbance at 532 nm was then determined. The standard curve is used to calculate the amount of thiobarbituric reactive chemicals produced, and the result is represented in nmole per mg protein [49].

The Sedlak and Lindsay approach was used to measure reduced glutathione (GSH) (1968). The sample was precipitated with trichloroacetic acid after being homogenized in 5 mL of infant KCl (1%). (TCA). The reaction mixture contains 2 mL of Tris buffer (pH 8.9), 0.5 mL of DTNB, and 0.5 mL of supernatant. At 412 nm, the solution was read [50].

Claiborne’s 1975 approach was used to measure the catalase activity. The reaction mixture, which had a final volume of 3 mL, was made up of 2 mL of phosphate buffer, 0.95 mL of H_2_O_2_ (20 mM), and 0.05 mL of sample PMS (10%). At intervals of 30 s, changes in absorbance were recorded at 240 nm. Using the molar extinction coefficient 43.6 M^−1^ CM^−1^, catalase activity was determined as nmole H_2_O_2_ consumed per minute per milligram protein [51].

Stevens et al. (2000) [52] approach was used to measure the superoxide dismutase (SOD) through measuring the auto-oxidation of epinephrine for 3 min at 480 nm wavelength. The enzymatic reaction of SOD was expressed in nmol epinephrine protected from oxidation/min/mg protein through using extinction co-efficient of 4.02 × 10^3^ M^−1^ CM^−1^.

### 4.6. Assessment of Inflammation and Apoptosis

Assessment of pro-inflammatory cytokines (IL-1β, and TNFα) and apoptosis (Caspase 3) were carried out as per the standard protocol of Abcam UK by using Elisa reader.

### 4.7. Histopathological Inspection

Formalin buffer solution (10%) was used to fix kidney tissues overnight, after which they were waxed and sectioned to 5 µm lengths. Hematoxylin and eosin (H and E) were used to stain the slides, and a 400× light microscope was used for the histopathological examination.

## 5. Conclusions

The current study concludes that sesamin (SM) is effective in attenuating oxidative stress and inflammatory cytokines associated with CP-induced nephrotoxicity in rats. No apparent adverse effects were noted during treatment with SM.

## Figures and Tables

**Figure 1 ijms-23-11615-f001:**
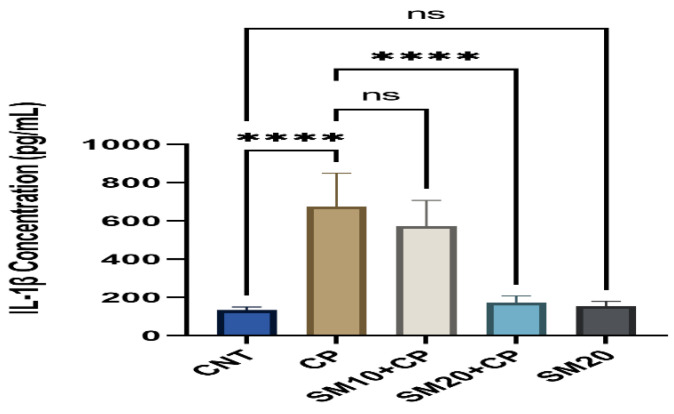
Protective effects of SM on renal inflammatory cytokine (IL-1β) level against CP-induced nephrotoxicity. Value denoted in mean ± SD (n = 6). **** *p* < 0.0001 vs. CNT, **** *p* < 0.0001 vs. CP and ^ns^
*p* > 0.05 vs. CNT, ^ns^
*p* > 0.05 SM10 vs. CP.

**Figure 2 ijms-23-11615-f002:**
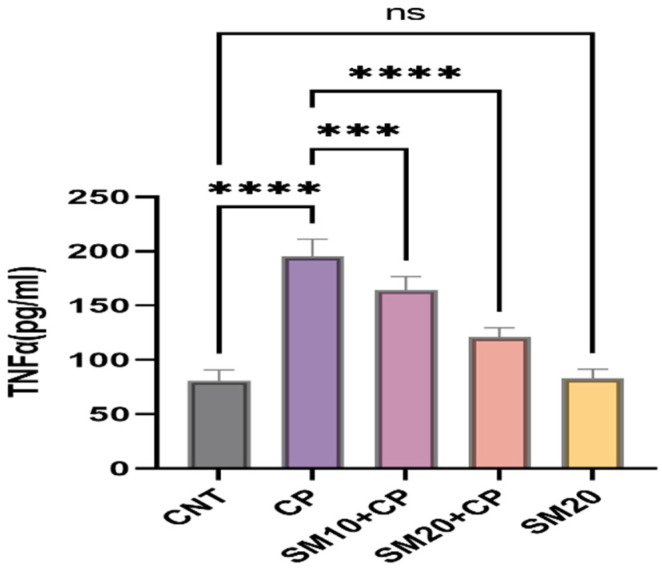
Protective effects of SM on renal inflammatory cytokine (TNFα) level against CP-dependent nephrotoxicity. Value denoted in mean ± SD (n = 6). **** *p* < 0.0001 vs. CNT, *** *p* < 0.001 vs. CP, **** *p* < 0.0001 vs CP and ^ns^
*p* > 0.05 vs. CNT,. Abbreviation; CNT: control, CP: cyclophosphamide, SM: sesamin, TNFα: tumor necrosis factor alpha.

**Figure 3 ijms-23-11615-f003:**
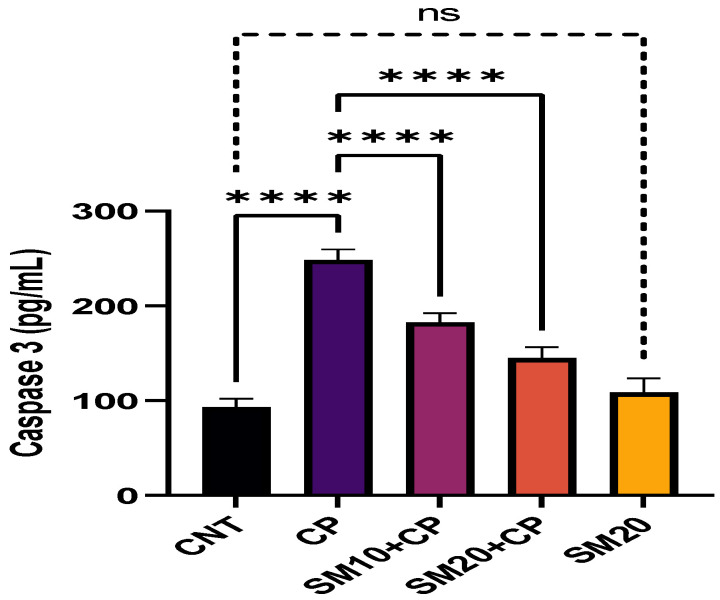
Protective effects of SM on renal apoptotic marker (Caspase 3) level against CP-induced nephrotoxicity. Value denoted in mean ± SD (n = 6). **** *p* < 0.0001 vs. CNT, **** *p* < 0.0001 vs. CP and ^ns^
*p* > 0.05 vs. CNT. Abbreviation; CNT: control, CP: cyclophosphamide, SM: sesamin.

**Figure 4 ijms-23-11615-f004:**
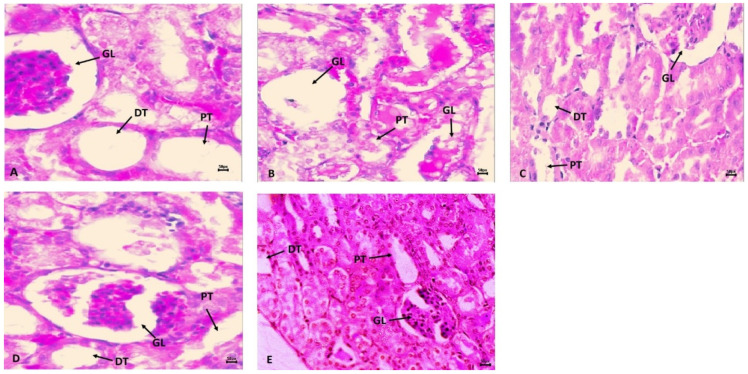
Protective effects of sesamin against cyclophosphamide induced renal toxicity. Hematoxylin and eosin (H and E) staining indicates (**A**) normal control (CNT) without any abnormality; (**B**) CP intoxication with clear abnormality such as necrosis in GL, PT, and DT. (**C**) SM10 + CP and (**D**) SM20 + CP is sesamin treatment with improvement in renal architect. (**E**) SM20 no significant changes in renal architect. Scale bar is 50 px.

**Table 1 ijms-23-11615-t001:** Effect of SM on serum markers against CP-induced nephrotoxicity.

Test Group	Blood Urea Nitrogen (mg/dL)	Uric Acid (mg/dL)	Creatinine (mg/dL)
CNT	18.66 ± 2.58	4.57 ± 1.18	0.81 ± 0.08
CP	50.16 ± 3.97 ^a^ (168.81%)	16.71 ± 2.64 ^b^ (265.64%)	2.06 ± 0.44 ^c^ (154.32%)
SM10 + CP	29.16 ± 4.71 ^d^ (−40.86%)	9.36 ± 1.66 ^e^ (−43.98%)	1.57 ± 0.78 ^f^ (−23.78%)
SM20 + CP	21.16 ± 2.79 ^g^ (56.93%)	5.63 ± 1.32 ^h^ (−66.30%)	0.91 ± 0.1 ^i^ (−55.82%)
SM20	19.00 ± 3.03 ^j^ (1.82%)	4.84 ± 1.47 ^k^ (5.90%)	0.85 ± 0.04 ^l^ (4.93%)

Data presented as mean ± standard deviation where n = 6, ^a^
*p* < 0.0001 vs. CNT, ^b^
*p* < 0.0001 vs. CP, ^c^
*p* < 0.0001 vs. CNT; ^d^
*p* < 0.0001 vs. CP, ^e^
*p* > 0.05 vs. CP, ^f^
*p* < 0.0001 vs. CP; ^g^
*p* < 0.0001 vs. CP, ^h^
*p* < 0.001 vs. CP, ^i^
*p* < 0.0001 vs. CP; ^j,k,l^
*p* > 0.05 vs. CNT.

**Table 2 ijms-23-11615-t002:** Effects of SM on oxidative stress against CP-induced renal toxicity.

Test Group	MDA (nmole/mg Tissue)	GSH (DTNB Conjugate Formed/mg Protein)	CAT (nmole of H_2_O_2_ Consumed/min/mg Protein	SOD (nmole Ephinephrine Protected from Oxidation/min/mg Protein
CNT	9.07 ± 1.37	21.95 ± 3.39	15.01 ± 2.41	33.89 ± 1.07
CP	27.3 ± 3.65 **	6.23 ± 1.25 ^##^	5.45 ± 1.30 ^a^	13.35 ± 3.45 ^$$^
SM10 + CP	15.66 ± 2.98 ***	12.89 ± 2.17 ^###^	10.74 ± 1.76 ^b^	25.42 ± 4.35 ^$$$^
SM20 + CP	10.50 ± 1.49 ***	19.89 ± 2.43 ^###^	12.98 ± 1.74 ^b^	33.22 ± 1.58 ^$$$^
SM20	10.12 ± 1.20 *	21.027 ± 2.25 ^#^	14.15 ± 2.65 ^c^	34.04 ± 3.55 ^$^

Data represent the protective effects of SM on oxidative stress markers (MDA, GSH, CAT and SOD) level against CP-induced nephrotoxicity. Data represented in mean ± SD (n = 6). * *p* > 0.05 vs. CNT, ** *p* < 0.0001 vs. CNT and *** *p* < 0.0001 vs. CP. ^#^
*p* > 0.05 vs. CNT, ^##^
*p* < 0.0001 vs. CNT, ^###^
*p* < 0.0001 vs. CP, ^c^
*p* > 0.05 vs. CNT, ^b^
*p* < 0.0001 vs. CP, ^a^
*p* < 0.0001 vs. CNT. ^$^
*p* > 0.05 vs. CNT, ^$$^
*p* < 0.0001 vs. CNT, ^$$$^
*p* < 0.0001 vs. CP. Abbreviation; MDA: malondialdehyde, CNT: control, CP: cyclophosphamide, GSH: glutathione, CAT: catalase, SOD: superoxide dismutase, SM: sesamin.

## Data Availability

The data presented in this study are available withing the article.

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
