# Peer review of "The Protective Effects of Sesamin against Cyclophosphamide-Induced Nephrotoxicity through Modulation of Oxidative Stress, Inflammatory-Cytokines and Apoptosis in Rats"

_ijms, 2022, doi:10.3390/ijms231911615_

Round 1

Reviewer 1 Report

·        Pg 4 Bowman’s space instead bowman’s space

·        Pg 4 The term “recovery” is not the best, because the treatment is a prophylactic one

·        Discussion: Inflammation markers instead of mediators of cytokines

·        Please present more clear the experimental design for each group, the route of administration, the time of administration, duration of the treatment

·        4.5. Assessment of toxicity markers replace by  Assessment of oxidative stress markers

·        Pg 7. H2O2 instead h2o2

·        4.7 Assessment of protein content. You have no results or interpretation for this parameter

·        References:  first reference (Newton HB) is not appropriate

Author Response

To The Editor

Good morning

I am thankful to you for your prompt response and fast reviewing process of our manuscript entitled “The Protective Effects of Sesamin against Cyclophosphamide-Induced Nephrotoxicity through Modulation of Oxidative Stress, Inflammatory- Cytokines and Apoptosis in Rats” and manuscript no ijms-1936123, I have gone through each comment raised by rewires and corrected in the revised manuscript accordingly. I have also tried to minimize the similarity of the whole manuscript, and now you will get acceptable and excited.

Reviewer-1 Reply

I appreciate the reviewer comments and corrected in the whole manuscript accordingly

  • Pg 4 Bowman’s space instead bowman’s space-

Reply: Corrected in Manuscript as suggested and highlighted in yellow color

  • Pg 4 The term “recovery” is not the best, because the treatment is a prophylactic one

Reply: The term recovery is replaced with regaining

  • Discussion: Inflammation markers instead of mediators of cytokines

      Reply- Replaced as suggested by respected reviewers

  • Please present more clear the experimental design for each group, the route of administration, the time of administration, duration of the treatment

Reply: I appreciate the reviewers comments and we have restructured the experimental design and focusing on the route of administration, duration of treatment etc and highlighted in yellow color

  • 4.5. Assessment of toxicity markers replace by  Assessment of oxidative stress markers

Reply: I have replaced the Assessment of toxicity markers as suggested by reviewers and highlighted.

  • Pg 7. H2O2 instead h2o2

      Reply: Corrected and highlighted as suggested by reviewer

  • 4.7 Assessment of protein content. You have no results or interpretation for this parameter

Reply: Thanks for your valuable comments and I have deleted the section from the manuscript to avoid any confusions.

  • References:  first reference (Newton HB) is not appropriate

Reply: I have replaced the Newton HB reference with more suitable reference and highlighted 1. Lameire N. Nephrotoxicity of recent anti-cancer agents. Clin Kidney J. 2014;7(1):11-22

Reviewer 2 Report

This work evaluated the protective of sesamin against cyclophosphamide induced renal toxicity. The positive control group is missing. The major comments are as below:

P1-line 3 in the 1st paragraph in introduction section: change depend to depending.

P1-line 4 in the 1st paragraph in introduction section: predicate structures are confused.

P1-line 5 in the 1st paragraph in introduction section: change drug to drugs.

P1-line 2 in the 2st paragraph in introduction section: which of the two metabolites?

P2-line 2 from the bottom in the last paragraph in introduction section: change encouraging to encourages.

Keep data retention digits same in tables.

P3-line 8: The subject and predicate are inconsistent.

P3-line 1 in ‘2.3. Inflammation and apoptosis’ section: change shows to showed.

P3-the 1st sentence in ‘2.4. Histopathology’ section: lack predicate.

P3-the last 2st sentence in ‘2.4. Histopathology’ section: delete the second ‘were’.

P5-line 7 in the 3rd paragraph in discussion section: change convert to converts.

P5-line 9 in the 3rd paragraph in discussion section: change interacting to interacts.

P5-line 4 in the 4rd paragraph in discussion section: change causes to caused.

P6-line 3: change suggests to suggested.

Author Response

Comment Reply to Reviewers

To The Editor

Good morning

I am thankful to you for your prompt response and fast reviewing process of our manuscript entitled “The Protective Effects of Sesamin against Cyclophosphamide-Induced Nephrotoxicity through Modulation of Oxidative Stress, Inflammatory- Cytokines and Apoptosis in Rats” and manuscript no ijms-1936123, I have gone through each comment raised by rewires and corrected in the revised manuscript accordingly. I have also tried to minimize the similarity of the whole manuscript, and now you will get accepted.

Reviewer-2 Reply

I appreciate the reviewer's comments and corrected the whole manuscript accordingly.

Comments and Suggestions for Authors

This work evaluated the protective of sesamin against cyclophosphamide induced renal toxicity. The positive control group is missing. The major comments are as below:

Reply: As a reviewer said, the positive control group is missing. I disagree with the respect that I have already taken a positive control group, and that is the SM20 group. Where we have given only a high dose of sesamin (SM20) to find the adverse effect compared to normal control. 

P1-line 3 in the 1st paragraph in introduction section: change depend to depending.

Reply: Changed as suggested by reviewers.

P1-line 4 in the 1st paragraph in the introduction section: predicate structures are confused.

Reply: Re structured this sentence to avoid confusion

P1-line 5 in the 1st paragraph in the introduction section: change drug to drugs.

Reply: corrected as suggested by the reviewers.

P1-line 2 in the 2st paragraph in introduction section: which of the two metabolites?

Reply: The two metabolites are named as  scarlein and phosphoramide mentioned in manuscript and highlighted

P2-line 2 from the bottom in the last paragraph in introduction section: change encouraging to encourages.

Reply: corrected as suggested by the reviewers

Keep data retention digits same in tables.

Reply: yes, agree and kept the same as it

P3-line 8: The subject and predicate are inconsistent.

Reply: Re-structured this sentence to make consistent

P3-line 1 in ‘2.3. Inflammation and apoptosis’ section: change shows to showed.

Reply: Changed as suggested by the reviewers and highlighted in yellow color

P3-the 1st sentence in ‘2.4. Histopathology’ section: lack predicate.

Reply: Histopathology section restructured and highlighted in yellow color

P3-the last 2st sentence in ‘2.4. Histopathology section: delete the second ‘were’.

Reply: Deleted “were” as suggested by the reviewers

P5-line 7 in the 3rd paragraph in discussion section: change convert to converts.

Reply: corrected as suggested by the reviewers.

P5-line 9 in the 3rd paragraph in discussion section: change interacting to interacts.

Reply: corrected as suggested by the reviewers and highlighted

P5-line 4 in the 4rd paragraph in discussion section: change causes to caused.

Reply: corrected as suggested by the reviewers

P6-line 3: change suggests to suggested.

Reply: corrected as suggested by the reviewers.

Round 2

Reviewer 2 Report

The authors has improved the manuscript and the work may be published in the Journal.